# Plasma Fibrinogen as a Predictor of Perioperative-Blood-Component Transfusion in Major-Nontraumatic-Orthopedic-Surgery Patients: A Cohort Study

**DOI:** 10.3390/diagnostics13050976

**Published:** 2023-03-04

**Authors:** Eugenio Pagnussatt Neto, Paula Daniele Lopes da Costa, Sanderland J. Tavares Gurgel, Paula Schmidt Azevedo, Norma S. Pinheiro Modolo, Paulo do Nascimento Junior

**Affiliations:** 1Department of Surgical Specialties and Anesthesiology, School of Medicine, Universidade Estadual Paulista (UNESP), Botucatu 18618-970, SP, Brazil; 2São Vicente de Paulo Hospital, Passo Fundo 99010-112, RS, Brazil; 3Department of Internal Medicine, School of Medicine, Universidade Estadual Paulista (UNESP), Botucatu 18618-970, SP, Brazil

**Keywords:** anesthesia, blood transfusion, blood-coagulation tests, fibrinogen, orthopedic surgeries

## Abstract

There is a trend towards increased perioperative bleeding in patients with plasma fibrinogen levels < 200 mg/dL^−1^. This study aimed to assess whether there is an association between preoperative fibrinogen levels and perioperative blood-product transfusion up to 48 h after major orthopedic surgery. This cohort study included 195 patients who underwent primary or revision hip arthroplasty for nontraumatic etiologies. Plasma fibrinogen, blood count, coagulation tests, and platelet count were measured preoperatively. A plasma fibrinogen level of 200 mg/dL^−1^ was the cutoff value used to predict blood transfusion. The mean (SD) plasma fibrinogen level was 325 (83) mg/dL^−1^. Only thirteen patients had levels < 200 mg/dL^−1^, and only one of them received a blood transfusion, with an absolute risk of 7.69% (1/13; 95%CI: 1.37–33.31%). Preoperative plasma fibrinogen levels were not associated with the need for blood transfusion (*p* = 0.745). The sensitivity and the positive predictive value of plasma fibrinogen < 200 mg/dL^−1^ as a predictor of blood transfusion were 4.17% (95%CI: 0.11–21.12%) and 7.69% (95%CI: 1.12–37.99%), respectively. Test accuracy was 82.05% (95%CI: 75.93–87.17%), but positive and negative likelihood ratios were poor. Therefore, preoperative plasma fibrinogen level in hip-arthroplasty patients was not associated with the need for blood-product transfusion.

## 1. Introduction

Fibrinogen, a 340 KDa glycoprotein comprising pairs of three polypeptide chains termed Aα, Bβ, and γ, is an essential factor in the coagulation cascade, due to its important role in platelet aggregation and the formation of an insoluble clot during the fibrin polymerization process [1,2]. Low levels of fibrinogen are found in hereditary disorders such as hypofibrinogenemia, afibrinogenemia, and dysfibrinogenemia, as well as in disseminated intravascular coagulation, fibrinolytic syndromes, and some liver diseases. In such conditions, surgical interventions require a precise hematologic management, balancing the risks of both bleeding and thromboembolic events [3]. Plasma fibrinogen levels < 200 mg/dL^−1^ may increase the risk of perioperative bleeding in obstetric patients (with a 100% predictive value for postpartum hemorrhage) [4], polytraumatized patients (mainly < 50 mg/dL^−1^) [5], and cardiac-surgery patients (three times higher when <100 mg/dL^−1^) [6].

The increase in life expectancy has led to more surgical procedures, such as large-joint orthopedic surgeries, which carry a high risk of thromboembolic events, as well as significant bleeding [7]. Orthopedic surgery often requires the administration of blood products, due to the large amount of blood loss associated with bone trauma and the difficulty in achieving hemostasis in bone tissue. The implementation of strategies to reduce the need for transfusions should start in the preoperative period, including risk identification and the improvement of intraoperative measures to reduce bleeding. Furthermore, the validation of risk-assessment models may also contribute to reducing bleeding-related complications and hospital costs [8].

Many risk models have been developed to estimate the risk of bleeding and blood transfusion, but most involve cardiac-surgery and trauma patients. Although the role of plasma fibrinogen in bleeding and transfusion prediction is not clearly defined, it has been included in most bleeding-risk models and has been shown to possibly identify patients at a higher risk of blood loss in cardiac surgeries, allowing anesthesiologists and surgeons to take additional measures to reduce bleeding and blood transfusion [9]. Therefore, the objective of this study was to determine whether there is a relationship between low preoperative levels of plasma fibrinogen and increased rates of blood-component transfusion in the perioperative period of elective orthopedic-hip surgery.

## 2. Materials and Methods

This prospective cohort study was conducted at Hospital São Vicente de Paulo, in the city of Passo Fundo, Brazil, after approval by the Hospital’s Graduate Research Commission on 26 March 2018, and the Universidade Estadual Paulista—Botucatu School of Medicine Research Ethics Committee (number 2.700.753; 4 June 2018). All participants provided written informed consent prior to inclusion. This study was conducted in accordance with the Strengthening the Reporting of Observational Studies in Epidemiology (STROBE) recommendations [10].

Patients aged 18 years or older of both sexes undergoing hip arthroplasty or surgical revision of nontraumatic hip arthroplasty between June 2018 and August 2021 were included in the study. Only patients whose preoperative plasma-fibrinogen level was measured using the Clauss method were included in the sample [11]. Patients diagnosed with liver cirrhosis or other active liver disease, thrombocytopenia <100,000 per mm^3^, and those on full or prophylactic anticoagulation, with the exception of acetylsalicylic acid, were excluded.

Laboratory tests such as blood count, platelet count, activated partial thromboplastin time, prothrombin time, international normalized ratio, and plasma fibrinogen were performed preoperatively, and the surgical team decided whether or not to perform a blood-product transfusion. The choice of anesthetic technique and whether or not to perform a blood-product transfusion was at the assistant anesthesiologist’s discretion. Data on anesthetic technique, antifibrinolytic therapy, operative time, blood transfusion, and referral to the postanesthesia-care unit or intensive care unit were collected intraoperatively. All procedures were performed by the same surgeon, to maintain technical uniformity. The patients were reassessed 24 h and 48 h after the procedure regarding the blood-product transfusion or referral to the intensive care unit. Finally, patient follow-up was conducted via telephone contact to assess 30- and 90-day survival.

### Statistics

Considering that the incidence of hypofibrinogenemia in patients at risk of surgical bleeding is 14% to 19.2% in trauma victims [5,12], 32% in liver-transplantation patients [13], and 23.8% to 45.1% in cardiovascular-surgery patients [14,15,16], the proportion of exposed patients (i.e., plasma fibrinogen < 200 mg/dL^−1^) was initially estimated at 30% for sample-size calculation. Since a 23%-to-54% probability of blood transfusion has been estimated in patients with hypofibrinogenemia [1,17], we estimated the risk of transfusion in patients with fibrinogen deficiency to be 50%. In contrast, blood transfusion in the unexposed group (i.e., fibrinogen > 200 mg/dL^−1^) was estimated at 20%, based on the 20% to 70% risk of transfusion in patients from the general population undergoing large-joint arthroplasty [18,19]. Therefore, considering a two-tailed α of 0.05 and a power of 80%, it was initially estimated that 103 participants would be needed to assess whether there is an association between preoperative plasma-fibrinogen levels and blood-product transfusion in orthopedic-hip surgery. After a preliminary data analysis from 103 patients, the magnitude of plasma fibrinogen < 200 mg/dL^−1^ in our sample was lower than that found in the literature. Thus, the sample size was adjusted according to the incidence of hypofibrinogenemia, reducing the estimate from 30% to 10% in relation to the proportion of exposed participants (fibrinogen < 200 mg/dL^−1^) in the first stage. The estimated risk of transfusion in patients with fibrinogen deficiency was maintained at 50%, as well as that of the unexposed group (20%). Considering the adjustments, a two-tailed α of 0.05, and a power of 80%, 189 patients were required to determine whether there is an association (*p* < 0.05) between preoperative plasma fibrinogen levels and perioperative blood-product transfusion in hip arthroplasty. To correct for follow-up losses, we recruited a total of 200 patients.

Descriptive data were presented as mean (SD) for quantitative variables and as absolute and relative frequency (%) for categorical variables. The chi-square test or Fisher’s exact test was used to compare categorical variables, whereas Student’s *t*-test or the nonparametric Mann–Whitney *U* test was used for numerical variables. A *p*-value < 0.05 was considered statistically significant. Because data collection involved diagnostic tests, the accuracy, likelihood ratios, sensitivity, specificity, and positive and negative predictive values of plasma-fibrinogen measurements were calculated as predictors of perioperative blood-product transfusion. The absolute and relative risks of blood-product transfusion were calculated in relation to preoperative plasma fibrinogen levels. Analyses were conducted using MedCalc (https://www.medcalc.org/, accessed on 7 April 2022) and IBM SPSS version 18.0.

## 3. Results

During the study period, 378 patients were evaluated for hip arthroplasties, either with primary surgeries or revision arthroplasties. Of these, the first 200 patients who met the inclusion criteria were enrolled. The surgery of five patients was cancelled due to lack of surgical material. The study flow diagram is presented in Figure 1.

Of the total sample, 24 (12.3%) patients received a blood-product transfusion at one or more moments during the perioperative period. Only one patient received packed red-blood cells during the preoperative period. In the intraoperative period, four patients received a transfusion of packed red-blood cells, and two also received fresh frozen plasma. In the first 24 h postoperatively, five patients received a transfusion of packed red-blood cells, of whom four were also transfused at other moments: one intraoperatively, and three within 24 h to 48 h postoperatively. Twenty patients received a transfusion of packed red-blood cells postoperatively, of whom four were previously transfused: one intraoperatively and three on the first postoperative day. No patient received cryoprecipitate transfusion. Thus, there was a predominance of blood transfusion between 24 h and 48 h after surgery (Figure 2).

The clinical-epidemiological characteristics of the patients and their association with blood-product transfusion are described in Table 1. No variable (age, sex, race, body mass index, or American Society of Anesthesiologists physical status) was associated with the need for blood transfusion (*p* > 0.05).

The mean (SD) plasma-fibrinogen level was 315 (83) mg/dL^−1^. There was no association between preoperative plasma fibrinogen and perioperative blood-product transfusion (*p* = 0.745). Preoperative blood counts (hemoglobin [*p* = 0.028] and hematocrit [*p* = 0.006]) were associated with higher transfusion rates, unlike platelet counts and any of the coagulation tests (Table 2). 

Thirteen patients had preoperative plasma fibrinogen levels < 200 mg/dL^−1^, of whom only one received a blood transfusion in the perioperative period. On the other hand, 182 patients had plasma fibrinogen levels ≥ 200 mg/dL^−1^, of whom 23 received blood-product transfusions. Thus, the relative risk of blood-product transfusion in patients with fibrinogen < 200 mg/dL^−1^ was 0.608 (95%CI: 0.089–4.158, *p* = 0.612). The results for plasma fibrinogen < 200 mg/dL^−1^ as a predictor of blood-product transfusion are presented in Table 3. Considering the intra- and postoperative variables, surgery type (*p* = 0.005), operative time (*p* = 0.001), and intensive-care-unit admission (*p* = 0.006) were associated with blood-product transfusion. In the postoperative period, hemoglobin and hematocrit levels were associated with a higher rate of packed red-blood cell transfusion (*p* < 0.001). Table 4 describes the outcomes in the intra- and postoperative periods. 

Of 195 patients in total, 158 (or their family members) were contacted to assess 30- and 90-day survival. A total of 37 participants/family members could not be contacted via telephone, which is why 30- and 90-day postoperative mortality were excluded from the statistical analysis. One of the contacted patients died, due to SARS-CoV-2 infection.

## 4. Discussion

According to the primary objective of this study, low preoperative plasma fibrinogen levels were not associated with higher perioperative blood-product transfusion rates in patients undergoing primary or revision hip arthroplasties.

Large-joint arthroplasties, especially of the hip and knee, are associated with excessive perioperative bleeding and a high demand for blood-product transfusion. Treatment should be tailored to ensure the balance of hemostasis between the replacement of clotting factors and thromboprophylaxis. Such demanding perioperative management was described in a patient with congenital coagulation disorder undergoing major hip surgery [20]. Restrictive strategies for blood-product transfusion have resulted in less need for blood transfusion. However, the reported incidence of blood-product transfusion in hip arthroplasties is still divergent, ranging from 5.6% to 70% [18,19,21,22,23]. In our study, only 12.3% of patients received a blood-product transfusion, showing a conservative trend regarding transfusion. Several factors may contribute to the difference in incidence, including different surgical techniques and surgical experience. Furthermore, undetected physiological-coagulation abnormalities may influence the risk of bleeding and the need for blood transfusion. In this setting, patients undergoing hip arthroplasty should be considered at a higher risk for bleeding and transfusion of blood products.

Of note, our population had no apparent risk factors for hypofibrinogenemia, unlike patients referred for cardiac surgery and trauma surgery. However, a study involving patients undergoing spinal arthrodesis with no apparent risks of hypofibrinogenemia reported a positive association between preoperative plasma fibrinogen levels and transfusion [24].

In this study, only 6.6% of patients had fibrinogen levels < 200 mg/dL^−1^. The mean patient age was 62 years, and most participants had controlled comorbidities and were overweight, which does not seem to be significantly related to fibrinogen changes. We measured plasma fibrinogen levels using the Clauss method, which is endorsed by the National Committee of Clinical Laboratory Standards [11]. In our study, although plasma fibrinogen measurement was not always performed in the same place, the assay was standardized between laboratories.

Most studies associating plasma fibrinogen levels with hemorrhagic events and transfusion in the perioperative period were performed in patients undergoing cardiac surgery, and showed conflicting results [25,26,27]. Studies of orthopedic surgeries with increased risk for blood loss, such as spinal osteotomies and joint arthroplasties, are lacking, and results involving fibrinogen as a prognostic test are controversial. Most of these studies involve patients undergoing spinal surgery, with a positive association between hypofibrinogenemia and increased risk of bleeding and transfusion [17,22,28]. In a prospective observational study that followed 245 patients undergoing spinal arthrodesis and hip or knee arthroplasties, a correlation was found between low fibrinogen levels and bleeding volume, or bleeding > 2 L, but only among patients undergoing spinal arthrodesis [22]. Because the methods for measuring intraoperative bleeding volume are imprecise, and there are no standardized definitions regarding surgical-bleeding levels, we did not include bleeding volume as an outcome in our study.

Regarding the validation of prognostic tests and their methodology, only a few studies investigating fibrinogen level as a predictor of blood-product transfusion or bleeding described test characteristics such as sensitivity, specificity, and predictive values. A study involving patients undergoing myocardial revascularization found a positive association between preoperative fibrinogen levels and increased postoperative bleeding, but the results showed low sensitivity and specificity (41.18% and 69.16%, respectively) for screening purposes [29]. Based on prognostic test analysis in orthopedic surgeries, evidence regarding the role of fibrinogen as a blood-transfusion biomarker are divergent. A retrospective study involving trauma patients analyzed the performance of plasma fibrinogen as a predictor of massive transfusion. The results showed a cutoff value of 211 mg/dL^−1^, with a sensitivity of 55.1% and a specificity of 78.6%, for identifying patients at risk [30]. A multicenter study with the same methodological design found a fibrinogen cutoff point of 190 mg/dL^−1^ for massive transfusion, but with a sensitivity of 78%, a specificity of 65%, and positive and negative predictive values of 34% and 81%, respectively [31]. In a prospective cohort of patients with pelvic fractures, lower systolic-blood pressure and fibrinogen levels were shown to be independent predictors of massive transfusion, the latter with high sensitivity and specificity (95.5% and 82.9%, respectively) [32]. 

The results of our study indicate that reduced preoperative plasma fibrinogen has low sensitivity for identifying patients at higher risk for blood-product transfusion, which compromises its performance as a biomarker for the risk of blood transfusion. These findings were probably due to our sample of nontraumatic-hip-surgery patients and because, in addition to fibrinogen, endothelial alterations, tissue-factor release, inflammatory response, and impaired fibrinolysis are some of the mechanisms responsible for perioperative hemostasis [7]. In contrast, the high specificity shows that the probability of blood transfusion in patients with normal fibrinogen is very low. 

Highly specific tests usually have a higher positive predictive value, due to the presence of fewer false-positive results [33]. This means that patients with low preoperative plasma fibrinogen levels would likely receive a transfusion. Paradoxically, our results showed a low positive predictive value and a high negative predictive value, suggesting that patients with normal plasma fibrinogen levels will not require blood-product transfusion. Because the prevalence of the predictor variable interferes with the predictive characteristics of the test [33], the lower prevalence of patients with fibrinogen levels below the cutoff point in our sample (6.6%) compared with the literature may have reduced the positive predictive value.

Although our results indicate good accuracy, we suggest caution in interpreting these data, since the sensitivity and positive predictive value were not expressed, leading to the conclusion that low preoperative plasma fibrinogen levels cannot be associated with a higher probability of blood transfusion. Furthermore, a good alternative for assessing the accuracy of a test is the likelihood ratio [34,35]. According to our sensitivity and specificity results, the positive likelihood ratio was 0.59 (95%CI: 0.08–4.36), which indicates null accuracy or a low diagnostic value [33]. The negative likelihood ratio was 1.03 (95%CI: 0.94–1.13), which indicates null accuracy or no diagnostic value [34], with minimal or no influence on the probability of transfusion. 

In accordance with previous data, no patient presented results below the reference values in the coagulation tests, and no significant relationship was found between the results of the coagulation tests (prothrombin time, activated partial thromboplastin time, international normalized ratio) and increased rate of blood-product transfusion [36,37,38]. However, these analyses are exploratory, since the sample size was not estimated to determine this relationship.

Regarding preoperative and postoperative hemoglobin and hematocrit measurements, our results showed that they were significantly associated with blood-product transfusion. These measurements within 48 h of surgery suggest that postoperative monitoring protocols would be beneficial. A validation study for surgical-care protocols in hip and knee arthroplasty showed that preoperative hemoglobin and hematocrit measurements can be considered good predictors of postoperative transfusion, and that postoperative control can be based on the patient’s clinical situation. Transfusion rates were higher in patients with low preoperative hemoglobin levels, those of an older age, those with aspirin use, and with other comorbidities, and the authors recommended monitoring hemoglobin for 1 to 3 days postoperatively [39].

In different surgical disciplines, the benefits of treating anemia and iron deficiency in the preoperative period are clearly shown. In orthopedic surgeries, treatment with intravenous iron and subcutaneous erythropoietin 1 to 3 days before surgery was associated with a reduction in the rate of packed red-blood-cell transfusions from 37% to 24%, and nosocomial infections from 12% to 8%. In patients with hip fractures, treatment of anemia and iron deficiency was associated with a reduction in mortality from 9.4% to 4.8% [40]. Even though the benefits of perioperative blood management are well established, this study was observational in nature, and did not propose any intervention, diagnosis, or therapeutics in relation to the occurrence of anemia in the preoperative period.

Surgical time and surgery type were also relevant factors for higher transfusion rates, with the number of transfused patients being proportionally higher in the revision-hip-arthroplasties group. Finally, despite the small number of patients referred to the intensive care unit, there was an association between referral and blood transfusion.

Study limitations include the fact that this was a single-center, observational study conducted in a private institution, which could limit the external validity of the results. Nonetheless, the institution is a center of excellence in major orthopedic surgery and trauma, and is up to date with current practices in hip surgery. Considering the initial low incidence of blood transfusion in our sample and the small number of patients with low preoperative plasma fibrinogen levels, we recalculated the sample size to adequately assess whether preoperative levels of plasma fibrinogen could predict the risk of blood-product transfusion, reducing the chances of a type-2 error [34,41].

There was also no specific algorithm for recommending blood transfusion in the perioperative period, since we guaranteed the autonomy of each anesthesiologist and surgical team regarding this decision. However, the fact that the same team performed all surgical procedures most likely minimized the impact of different techniques on bleeding. Given the lack of conceptual definitions of surgical bleeding and the imprecision of bleeding-volume measurement, this variable was not considered in the analysis.

For the purposes of statistical analysis, we established a plasma-fibrinogen cutoff value to categorize the sample as normal (≥200 mg/dL^−1^) or abnormal (<200 mg/dL^−1^). This value was determined according to the literature, which describes normal values ranging from 200 to 400 mg/dL^−1^ [42], although different studies have chosen to analyze the plasma-fibrinogen cutoff value when analyzing the relationship between plasma fibrinogen levels and bleeding or blood-product transfusion.

## 5. Conclusions

Plasma fibrinogen levels prior to hip arthroplasty or revision surgery are not an effective biomarker of increased risk for blood-product transfusion. Particularly, levels < 200 mg/dL^−1^ have low sensitivity and low positive predictive value for packed red-blood-cell transfusion. There is no risk model for predicting blood-product transfusion in major orthopedic surgeries such as hip arthroplasties. According to our analysis, it is unnecessary to measure plasma fibrinogen as part of the preoperative assessment of patients undergoing hip arthroplasty or revision surgery to evaluate the perioperative risk of blood-product transfusion.

## Figures and Tables

**Figure 1 diagnostics-13-00976-f001:**
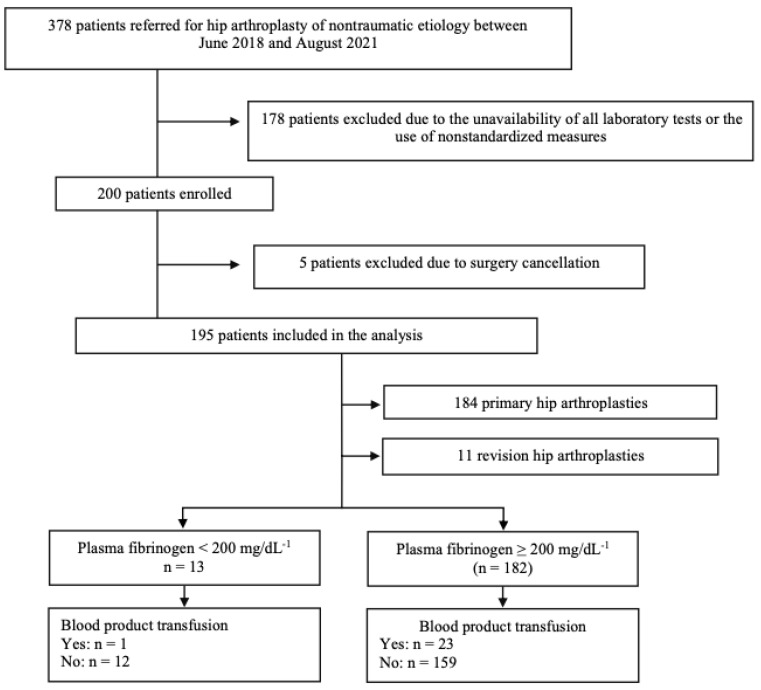
Study flow diagram.

**Figure 2 diagnostics-13-00976-f002:**
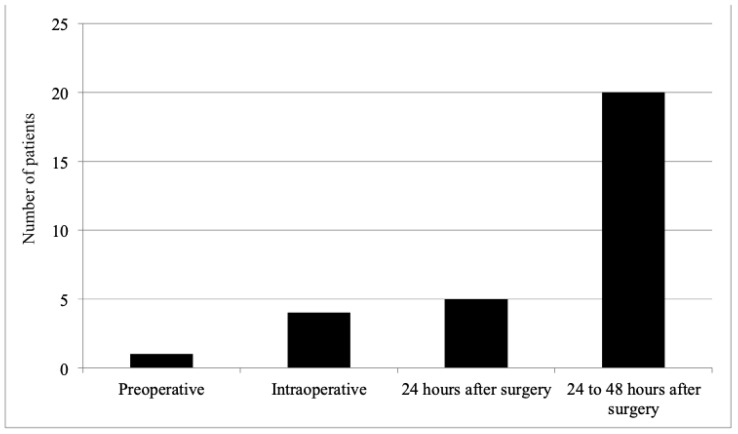
Frequency of blood-product transfusion at different moments of perioperative follow-up.

**Table 1 diagnostics-13-00976-t001:** Comparison of clinical-epidemiological characteristics between transfused and non-transfused patients.

			Blood-Product Transfusion		
Clinical-Epidemiological Characteristics		Total (*n* = 195)	No (*n* = 171)	Yes (*n* = 24)	*p*-Value
Age (years)		62.2 (11.2)	61.7 (11.3)	65.6 (10.3)	0.119 ‡
Sex	Female	91 (46.7%)	76 (83.5%)	15 (16.5%)	
	Male	104 (53.3%)	95 (91.3%)	9 (8.7%)	0.126 §
Race	White	194 (99.5%)	170 (87.6%)	24 (12.4%)	
	Mixed	1 (0.5%)	1 (100.0%)	0 (0.0%)	1.000 §
BMI *		27.7 (4.4)	27.9 (4.5)	26.8 (4.2)	0.259 ¶
ASA-PS †	1	35 (17.9%)	31 (88.6%)	4 (11.4%)	
	2	148 (75.9%)	132 (89.2%)	16 (10.8%)	
	3	12 (6.2%)	8 (66.7%)	4 (33.3%)	0.073 **

Data presented as mean (SD) for quantitative variables and as frequency (%) for categorical variables. * BMI: body mass index; † ASA-PS: American Society of Anesthesiologists—Physical Status. ‡ Mann–Whitney *U* test; § Fisher’s exact test; ¶ Student’s *t*-test; ** Chi-square test.

**Table 2 diagnostics-13-00976-t002:** Comparison of preoperative laboratory test results between transfused and non-transfused patients.

		Blood-Product Transfusion			
Preoperative Laboratory Tests		Total (*n* = 195)	No. (*n* = 171)	Yes (*n* = 24)	*p*-Value
Fibrinogen (mg/dL^−1^)		315 (83)	316 (83)	310 (79)	0.745 §
Hemoglobin (g/dL^−1^)		13.9 (1.3)	14.0 (1.2)	13.1 (1.8)	0.028 §
Hematocrit (%)		41.6 (4.1)	42.1 (3.9)	39.1 (5.1)	0.006 ¶
Platelets (count)		230,756 (61,063)	229,371 (61,298)	240,625 (59,689)	0.327 ¶
Coagulation	PT * (seconds)	12.2 (1.5)	12.1 (1.5)	12.7 (1.5)	0.093 ¶
	APTT † (seconds)	28.7 (6.5)	28.8 (6.7)	27.4 (4.8)	0.291 ¶
	INR ‡	1.0 (0.1)	1.0 (0.1)	1.1 (0.1)	0.449 ¶

Data presented as mean (SD). * PT: prothrombin time; † APTT: activated partial thromboplastin time; ‡ INR: International Normalized Ratio. § Student’s *t*-test; ¶ Mann-Whitney *U* test.

**Table 3 diagnostics-13-00976-t003:** Performance and prognostic capacity of plasma fibrinogen test as a predictor of blood-component transfusion.

Test Feature	Results *
Sensitivity	4.17% (95%CI: 0.11–21.12%)
Specificity	92.98% (95%CI: 88.06–96.32%)
Positive predictive value	7.69% (95%CI: 1.12–37.99%)
Negative predictive value	87.36% (95%CI: 86.30–88.35%)
Accuracy	82.05% (95%CI: 75.93–87.17%)
Positive likelihood ratio	0.59 (95%CI: 0.08–4.36)
Negative likelihood ratio	1.03 (95%CI: 0.94–1.13)
Absolute risk (fibrinogen < 200 mg/dL^−1^) (A)	7.69% (95%CI: 1.37–33.31%)
Absolute risk (fibrinogen ≥ 200 mg/dL^−1^) (B)	12.64% (95%CI: 8.57–18.25%)
Relative risk (A/B)	0.608 (95%CI: 0.089–4.158)

***** Values with respective 95%CIs are shown.

**Table 4 diagnostics-13-00976-t004:** Comparison of intraoperative and postoperative variables between transfused and non-transfused patients.

		Blood-Product Transfusion			
Variable		Total (*n* = 195)	No. (*n* = 171)	Yes (*n* = 24)	*p*-Value
Surgical technique (*n*)	THA *	184 (94.4%)	165 (89.7%)	19 (10.3%)	
	Revision THA	11 (5.6%)	6 (54.5%)	5 (45.5%)	0.005 §
Operative time (minutes)		120.2 (30.3)	117.0 (26.7)	143.1 (42.8)	0.001 ¶
Anesthetic technique (*n*)	Neuraxial	188 (96.4%)	165 (87.8%)	23 (12.2%)	
	General	5 (2.6%)	4 (80.0%)	1 (20.0%)	
	Other	2 (1.0%)	2 (100.0%)	0 (0.0%)	0.757 **
Antifibrinolytic therapy (*n*)	Yes, intravenous	188 (96.4%)	165 (87.8%)	23 (12.2%)	
	Yes, intra-articular	4 (2.1%)	3 (75.0%)	1 (25.0%)	
	No	3 (1.5%)	3 (100.0%)	0 (0.0%)	0.601 **
Hospitalization (*n*)	ICU †	4 (2.1%)	1 (25.0%)	3 (75.0%)	
	PACU ‡	191 (97.9%)	170 (89.0%)	21 (11.0%)	0.006 §
Postoperative hemoglobin (g/dL^−1^)		10.5 (1.5)	10.8 (1.3)	8.61 (1.6)	<0.001 ¶
Postoperative hematocrit (%)		31.5 (4.3)	32.2 (3.9)	26.3 (3.7)	<0.001 ††

Data presented as mean (SD) for quantitative variables and as frequency (%) for categorical variables. * THA: total hip arthroplasty; † ICU: intensive care unit; ‡ PACU: post-anesthesia care unit. § Fisher’s exact test; ¶ Mann–Whitney *U* test; ** Chi-square test; †† Student’s *t*-test.

## Data Availability

Data supporting reported results can be found at https://data.mendeley.com/datasets/brmtb5znhs, accessed on 14 October 2022.

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
