# Peer review of "Plasma Fibrinogen as a Predictor of Perioperative-Blood-Component Transfusion in Major-Nontraumatic-Orthopedic-Surgery Patients: A Cohort Study"

_diagnostics, 2023, doi:10.3390/diagnostics13050976_

Round 1

Reviewer 1 Report

The authors describe a very interesting original data, which is focused on plasma fibrinogen as a predictor of perioperative blood transfusion in orthopaedic surgery. The scope of the manuscript is adequate, the authors do well to focus on on the given issue.The authors do a solid job. The text is supported by several pictures and a table that complement the content of the manuscript. The most important finding was that the preoperative plasma fibrinogen level in hip arthroplasty patients was not related to the need for transfusion of blood products.

Major comments:

- Introduction-  Page 1, lines 38-40: The authors should describe the fibrinogen molecule in more detail. We think it is appropriate to add that: Fibrinogen is a 340 kDa glycoprotein comprising pairs of three polypeptide chains termed Aα, Bβ, and γ. It is appropriate to cite the manuscript in which it was written: Fibrinogen Martin: A Novel Mutation in FGB (Gln180Stop) Causing Congenital Afibrinogenemia. Semin Thromb Hemost. 2016 Jun;42(4):455-8. doi: 10.1055/s-0036-1581104.

- Page 1 lines 40-43: The authors should state that even a fibrinogen level of 0.5 g/l is very necessary in patients with a congenital fibrinogen disorder with a history of serious haemorrhage: It is appropriate to cite the manuscript in which it was written: Yes or no for secondary prophylaxis in afibrinogenemia? Blood Coagul Fibrinolysis. 2015 Dec;26(8):978-80. doi: 10.1097/MBC.0000000000000392.

- Discussion -  lines 197-207 In orthopedic surgery, it is necessary to think about hemostatic balance. The recommendation to tailor treatment to ensure a hemostasis balance between the replacement of clotting factor (fibrinogen concentrate) and thromboprophylaxis. Such demanding perioperative management was in orthopedic surgery for congenital coagulopathy, where it is important to think that increased fibrinogen substitution may have an increased risk of thrombosisIt is appropriate to state the following manuscript: Perioperative Coagulation Management in a Patient with Congenital Afibrinogenemia during Revision Total Hip Arthroplasty. Semin Thromb Hemost. 2016 Sep;42(6):689-92. doi: 10.1055/s-0036-1585079.

Tables and figures in the text are very clearly written.

I have to say that with these 38 references of which less than half are references from the last 5 years.

Author Response

REVIEWER 01

Introduction - Page 1, lines 38-40: The authors should describe the fibrinogen molecule in more detail. We think it is appropriate to add that: Fibrinogen is a 340 kDa glycoprotein comprising pairs of three polypeptide chains termed Aα, Bβ, and γ. It is appropriate to cite the manuscript in which it was written: Fibrinogen Martin: A Novel Mutation in FGB (Gln180Stop) Causing Congenital Afibrinogenemia. Semin Thromb Hemost. 2016 Jun;42(4):455-8. doi: 10.1055/s-0036-1581104.

Response 01

We included the suggested information and reference in the first sentence of Introduction.

We changed “Fibrinogen is an essential factor in the coagulation cascade due to its important role in platelet aggregation and the formation of an insoluble clot during the fibrin polymerization process [1].” to “Fibrinogen, a 340 KDa glycoprotein comprising pairs of three polypeptide chains termed Aα, Bβ, and γ, is an essential factor in the coagulation cascade due to its important role in platelet aggregation and the formation of an insoluble clot during the fibrin polymerization process [1,2]”.

Page 1 lines 40-43: The authors should state that even a fibrinogen level of 0.5 g/l is very necessary in patients with a congenital fibrinogen disorder with a history of serious haemorrhage: It is appropriate to cite the manuscript in which it was written: Yes or no for secondary prophylaxis in afibrinogenemia? Blood Coagul Fibrinolysis. 2015 Dec;26(8):978-80. doi: 10.1097/MBC.0000000000000392.

Response 02

We included the suggested information and reference in the second sentence of Introduction. However, we wrote it in the context of the surgical scenario, as the reference suggested by the reviewer mentions a clinical situation (afibrinogenemia) that also had to be managed for a surgical procedure.

We wrote: “Low levels of fibrinogen are found in hereditary disorders such as hypofibrinogenemia, afibrinogenemia, and dysfibrinogenemia, as well as in disseminated intravascular coagulation, fibrinolytic syndromes, and some liver diseases. In such conditions, surgical interventions require a precise hematologic management, balancing the risks of both bleeding and thromboembolic events [3].” Discussion - lines 197-207 In orthopedic surgery, it is necessary to think about hemostatic balance. The recommendation to tailor treatment to ensure a hemostasis balance between the replacement of clotting factor (fibrinogen concentrate) and thromboprophylaxis. Such demanding perioperative management was in orthopedic surgery for congenital coagulopathy, where it is important to think that increased fibrinogen substitution may have an increased risk of thrombosisIt is appropriate to state the following manuscript: Perioperative Coagulation Management in a Patient with Congenital Afibrinogenemia during Revision Total Hip Arthroplasty. Semin Thromb Hemost. 2016 Sep;42(6):689-92. doi: 10.1055/s-0036-1585079.

Response 03

We included the sentences suggested by the reviewer as well as the suggested reference in the second paragraph of Discussion.

We changed “Large joint arthroplasties, especially of the hip and knee, are associated with excessive perioperative bleeding and a high demand for blood product transfusion.” to “Large joint arthroplasties, especially of the hip and knee, are associated with excessive perioperative bleeding and a high demand for blood product transfusion. Treatment should be tailored to ensure the balance of hemostasis between the replacement of clotting factors and thromboprophylaxis. Such demanding perioperative management was described in a patient with congenital coagulation disorder undergoing major hip surgery [20].”

 - I have to say that with these 38 references of which less than half are references from the last 5 years.

Response 04

We agree with the reviewer regarding the number of references that are older than 5 years. Nonetheless, they provide solid evidence to our manuscript, and some of them have information on physiology and hemostasis that we consider relevant. We chose these references because information or new information about the role of fibrinogen as a biomarker for perioperative transfusion is scarce in the literature.

Reviewer 2 Report

Thank you for giving me the possibility to review the manuscript "Plasma fibrinogen as a predictor of perioperative blood components transfusion in major non-traumatic orthopaedic surgery patients". 

The paper deals with a very interesting topic in orthopaedic surgery, however, before considering it for publication in Diagnostics, the following points should be addressed:

1 INTRODUCTION: line 58: please clearly state the aim of the study

2 report preoperative ferritin and transferrin concentrations

3 assess any correlation between preoperative ferritin and transferrin concentration and postoperative Hb

4 please state if you have used a  preoperative blood management optimization procedure and detail it

Author Response

REVIEWER 02

  1. Introduction: line 58: please clearly state the aim of the study

Response 01

We stated the objective of the study in the last paragraph of Introduction: “Therefore, the objective of this study was to determine whether there is a relationship between low preoperative levels of plasma fibrinogen and increased rates of blood component transfusion in the perioperative period of elective orthopedic hip surgery.”

  1. Report preoperative ferritin and transferrin concentrations:
  2. Assess any correlation between preoperative ferritin and transferrin concentration and postoperative Hb:
  3. Please state if you have used a preoperative blood management optimization procedure and detail it:

Response 02-03-04

We made comments about perioperative blood management throughout the manuscript. However, we do not have information on preoperative ferritin and transferrin concentrations or any measurement regarding perioperative blood management. This is beyond the scope of our observational study.

In the last page of Discussion, we included the following information, as well as a new reference: “In different surgical disciplines, the benefits of treating anemia and iron deficiency in the preoperative period are clearly shown. In orthopedic surgeries, treatment with intravenous iron and subcutaneous erythropoietin 1 to 3 days before surgery was associated with a reduction in the rate of packed red blood cell transfusions from 37% to 24% and nosocomial infections from 12% to 8%. In patients with hip fractures, treatment of anemia and iron deficiency was associated with a reduction in mortality from 9.4% to 4.8% [40]. Even though the benefits of perioperative blood management are well established, this study was observational by nature and did not propose any intervention, diagnosis, or therapeutics in relation to the occurrence of anemia in the preoperative period.”

We emphasize that English language has been proofread by a professional reviewer and we hope to be able to publish in such a distinguished journal.

Round 2

Reviewer 1 Report

The presented manuscript has been corrected in response to the suggestions. The authors have followed the recommendations of the reviewer. After the revision, the provided data and addition of the results became more clear. I would like to thank the authors for resubmitting the manuscript and explaining the obscure points from the previous version.